# Occupational exposure to suicide: A review of research on the experiences of mental health professionals and first responders

Renan Lopes de Lyra[ID]*, Sarah K. McKenzie[o], Susanna Every-Palmer[ID][o], Gabrielle Jenkin[o]

Department of Psychological Medicine, University of Otago, Wellington, New Zealand

[o] These authors contributed equally to this work.
* renan.lyra@postgrad.otago.ac.nz

## Abstract

Exposure to suicide is a major factor for suicidality. Mental health professionals and first responders are often exposed to suicide while on-duty. The objective of this scoping review is to describe the state of current research on exposure to suicide among mental health professionals and first responders, focusing on the prevalence and impact of exposure to suicide, and to identify current gaps in the literature. We searched MEDLINE, Scopus, PsychNET, and Web of Science and identified 25 eligible papers. Between 31.5–95.0% of professionals had been exposed to suicide. Exposure to suicide had impacts on personal life, professional life, and mental health; and caused emotional distress. There was little research investigating exposure to suicide among police officers, firefighters, and paramedics. More research existed on mental health professionals, but none assessed exposure to suicide as a risk for suicide amongst this group. The review concludes that exposure to suicide is distressing for mental health professionals, and likely to be for first responder however, more research on these groups, especially paramedics, is required.

## Background

Suicide is a global health and social issue, with an estimated 800,000 people dying by suicide each year [1]. The social and psychological costs of suicide are high. For every suicide, it has been estimated that between six and 20 people, usually family members and acquaintances of those who died, are adversely affected psychologically and emotionally [1, 2]. This phenomenon has been described as 'exposure to suicide'. It is the impact of exposure to suicide on these two professional groups–mental health professionals and first responders–that underpins this scoping review.

In this review, we are particularly interested in two groups of people, who because of their occupations, have greater exposure to suicide than the rest of the population; mental health professionals and first responders [3, 4]. As well as having greater than usual exposure to suicide, these groups are distinct from bereaved family and friends due to the professional nature of their relationship with the person who died by suicide. Mental health professionals and first

**Data Availability Statement:** All relevant data are within the manuscript and its Supporting Information files.

**Funding:** This study was supported by University of Otago in the form of PhD Scholarship awarded to RLdL (Student number 5953060). The funder had no role in study design, data collection and analysis, decision to publish, or preparation of the manuscript.

**Competing interests:** The authors have declared that no competing interests exist.

responders, will in most cases, be unrelated to the person who died and be unlikely to have had a social or personal relationship with them. Yet these two groups of professionals are distinct from each other in that mental health professionals will often have established a therapeutic alliance with the person who died by suicide, while first responders will usually be unknown to the person prior to their suicide. Because the nature of the relationship with the deceased by suicide is different for mental health professionals than for first responders, we might expect the impacts of the exposure to suicide on these two groups to be different.

Mental health professionals (especially psychiatrists) and first responders themselves also have high rates of suicide [5, 6]. One of the explanations for the higher rates of suicides among these professions is their higher levels of occupational-related psychological distress, and for first responders, work-related Post Traumatic Stress Disorder (PTSD) [7]. In the literature, psychological distress is often talked about as vicarious trauma, or compassion fatigue [8]. Vicarious trauma (VT), defined as a disruption of perception and thoughts caused by the transmission of traumatic experiences when cumulatively witnessing disturbing clinical material or due to work-related emotional toll [8–11], can have both *personal* and *professional* impacts. On a personal level, VT is characterized by sleep disturbances, feelings of hopelessness, and disconnection from others [8].

One of the contributors to higher levels of psychological distress and VT among mental health professionals and first responders may be their higher *exposure to suicide*. A recent systematic review by [6] suggests a link between the psychological distress experienced in these occupations and higher rates of suicides. Attending suicide calls is one of the critical incidents ambulance personnel describe as leading to an emotional toll and therefore contributes to their mental health and well-being [12].

Exposure to suicide can affect first responders and mental health professionals personally and impact their professional practice. The changes in professional practice after a patient's suicide include heightened awareness of suicide risk, reduction in confidence, and more restrictive practices, including, for nurses, increasing the level of observations and detention [13, 14]. The management of suicidal behavior in patients can present as increased vigilance when assessing suicidality, or the systematic referral of a patient to another colleague (for a second evaluation), or avoidance of treating patients at risk of suicide altogether [13, 15].

Other professional implications include the questioning of confidence in career choice, which may impact occupational retention [16, 17]. In respect of re-assessing career choices, questioning one's career choice is not limited to health professionals, and has been observed in other professions that deal with emotionally sensitive content, for example, suicide researchers [9, 10].

For first responders, emotional impacts include high emotional labor (the process of controlling feelings according to organizational and duty demands), which has been found to be associated with increased suicidal ideation among firefighters [7]. This group of professionals is often those who first attend a suspected suicide scene, which has been described as one of the most critical and traumatic situations first responders may encounter during their work [18]. There is little discussion in the literature of strategies for managing psychological distress from exposure to suicide amongst first responders as a professional group.

While the research to date has provided some insights into the complex impacts of exposure to suicide on mental health professionals, there has been limited research on the impact of exposure to suicide on first responders. Furthermore, this literature has not been reviewed and examined together as phenomena of occupation-based exposure to suicide and its impacts. This scoping review aims to describe the current state of research in this field. The specific questions guiding this literature review are (i) what is known regarding the prevalence of exposure to suicide among mental health professionals (MHP) and first responders? (ii) what are

the personal and professional impacts of exposure to suicide on mental health professionals and first responders? and, (iii) what are the gaps in the literature on exposure to suicide among first responders and mental health professionals?

## Methods

A narrative scoping review was chosen due to the heterogeneity, of research, quantitative and qualitative, exploring exposure to suicide amongst MHP and first responders. Scoping reviews are similar to systematic reviews, but with different purposes, like to identify available evidence in a given field, to clarify conceptions, to exam how research is conducted in a certain topic, and to analyses and identification of research gaps [19]. Therefore, this method was the best fit for our research.

### Search strategy

**Databases.**   Eligible studies were identified through searches in four academic databases (MEDLINE, Scopus, PsychNET, and Web of Science).

**Search terms.**   Two separate searches were conducted using different search terms for each professional group. For MHP, the following search terms were applied: "Psychology, Clinical Psychology, Psychotherapy, Health Personnel, Psychiatry, Health Professionals" using Booleans and MeSH terms, and "suicide, grief, bereavement, exposure, client suicide, patient suicide" and Booleans. For first responders, the following search terms were applied: "firefighters, police officers, emergency responders, paramedics" using Booleans and MeSH terms, and "suicide, grief, bereavement, exposure" and Booleans (complete search strategy on request from authors).

**Eligibility criteria.**   The inclusion criteria comprised of peer-reviewed research published in English in the last ten years (from June 2010 to June 2020) that examined the prevalence of exposure to suicide among MHP and/or first responders and/or its impacts on these groups.

Non-primary research articles (discussions, editorials, protocol papers, reviews, and theoretical papers) were excluded. Articles that examined exposure to suicide solely focused on instrument validation were also excluded as they reported only psychometric data.

**Data chart.**   The eligible papers were read full-text and data was charted independently by the first author with contributions from the other three authors. The data extraction described the authorship, year, country, population, sample size, measured aspects, reported emotional impacts, reported professional impacts, exposure to suicide prevalence, reported support sought, reported mental health outcomes, other relevant findings reported, limitations reported, and instruments used.

**Study selection.**   We conducted a title and abstract review on the 381 eligible papers retrieved. to exclude studies not meeting our eligibility criteria. In total, 53 articles were allocated for full-text review, and of these we excluded 28 that did not meet our eligibility criteria, leaving 25 papers for full review (see Fig 1).

## Results

### Quality appraisal

The quality of included studies was assessed using a modified version of the CASP tool (Critical Appraisal Skills Programme UK, [20]) for qualitative and quantitative research. The mean score for the quality of qualitative research (where 10 is very good and 0 is poor) was 7.5 out of 10. One important and prominent weaknesses of the qualitative research affecting the validity of findings were the lack of reporting on researcher reflexivity, i.e. consideration of the impact

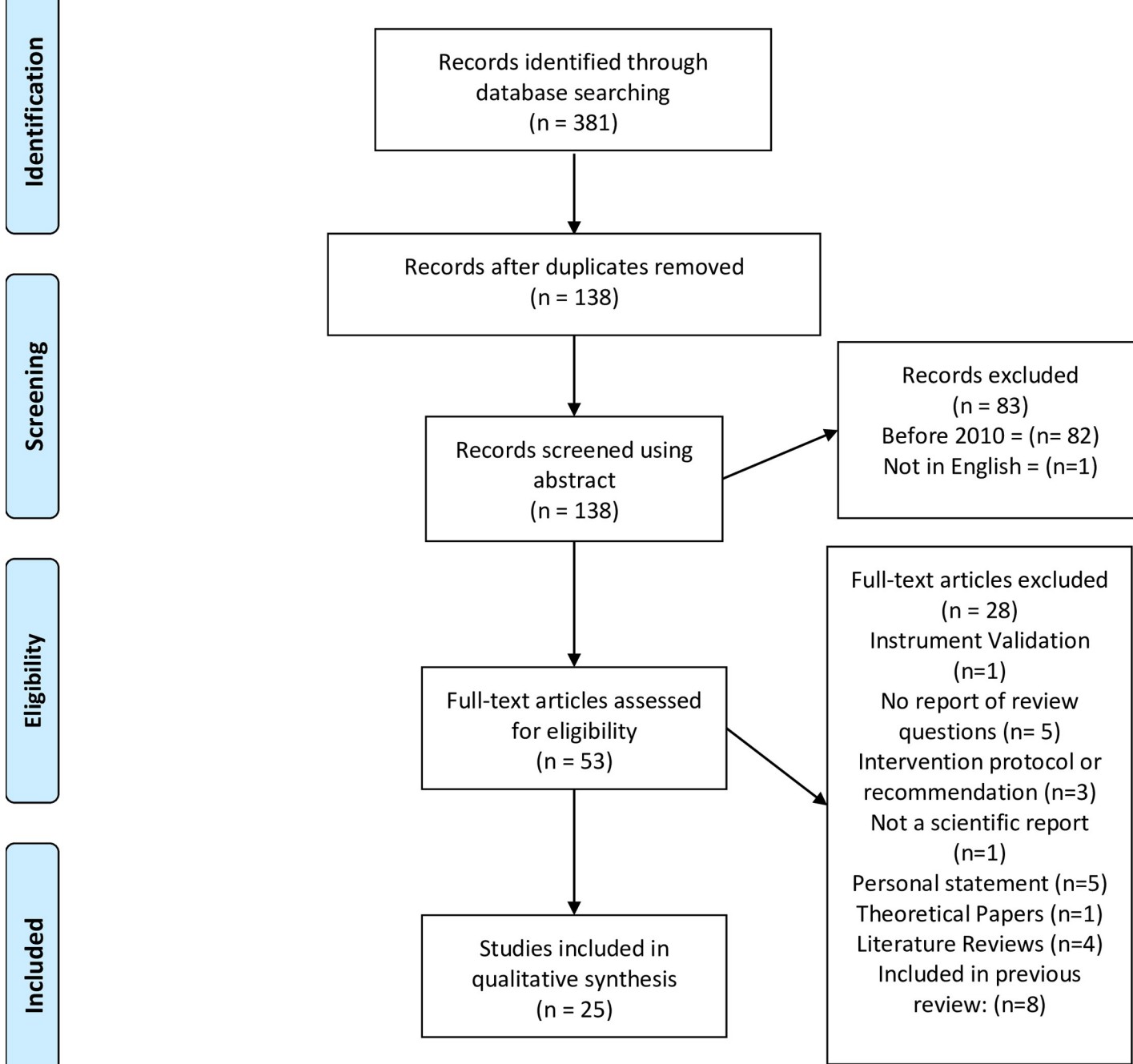

**Fig 1. Prisma chart flow.** Paper selection and eligibility for the review.

of researchers' role and characteristics resulting in potential bias or unconsidered influence on data collection and analysis. The mean score for the quality of quantitative research was 4.85 out of 7. Key weaknesses of the quantitative research affecting validity and generalizability included the use of non-validated instruments/measures for assessing mental health impacts (symptoms and emotional reactions, use of purposive sampling method (not representative), and small samples that lacked sufficient power.

## Literature characteristics

The majority of the 25 eligible papers (n = 15) were quantitative, four were qualitative methods and six were both. Seven papers assessed suicide impacts among all mental health professionals (MHPs) [21–27]. A total of 11 papers assessed suicide impacts among psychiatrists and psychologists only [4, 28–37], one assessed the suicide impact on nurses [38] and one assessed exposure to suicide reactions among counselors and social workers [39]. Only five papers addressed exposure to suicide among first responders (firefighters and police) [3, 40–43]. The findings of each paper are summarized on the Table 1.

## Mental health professionals

The MHPs group includes psychologists, psychiatrists, social workers, and nurses. Of the seven papers, four reported the prevalence of exposure to suicide among MHPs [21, 23–25]. The prevalence of exposure to at least one suicide, ranged from 32% to 46.6% with professionals experiencing an average of around 4 suicides throughout their career.

The emotional reactions described by these professionals included sadness, shock, surprise, feelings of blame, hopelessness, guilt, self-doubt, grief, and anger. Their professional reactions ranged from increased awareness of suicide risk, reduced professional confidence, fear of publicity and litigation, increased referrals to psychiatrists, and sadness at work. The type of support received by these MHPs in their day-to-day work was described as informal peer-support, including debriefing. Only two papers measured the mental health outcomes in MHPs following a patient's suicide, with burnout and PTSD reported as significant adverse outcomes [21, 27].

The setting of exposure to suicide for MHPs was also an important factor. Mental health facilities are designed to provide intensive care for psychiatric inpatients, yet, suicide is still common in these settings [22]. The suicide of an inpatient can be even more distressing, as it occurs in the workplace where MHP are both treating clinicians and first responders. Dransart, Gutjahr [27] findings suggest that professionals working in institutions in which patients require more care (e.g., aged care facilities) have an increased risk of being impacted by suicides as they perceive themselves more responsible for the patient than professionals in other hospital settings.

In terms of MHP sociodemographic factors, Draper et al. (2014) reported that being female, aged under 50 years, and having either less than five years of professional practice or between 11 and 20 years was significantly associated with changes in professional practice following a client suicide–e.g., hypervigilance of potential suicide cues. Similarly, being female, aged under 50 years, and having less than five years of practice was significantly associated with personal impacts, such as higher levels of emotional distress, sadness, and shock.

These findings suggest that both young and very experienced MHP's are equally vulnerable to exposure to suicide impacts. They also highlight the need for future research looking at gender differences in the impact of exposure to suicide on MHPs.

## Psychologists and psychiatrists

Of the eleven papers examining the impacts of exposure to suicide, eight reported the prevalence of patient's suicide focused on psychologists and psychiatrists, with 31.5–92% of these professionals exposed to at least one client suicide [4, 28–31, 34–36]. Moreover, Gibbons et al. (2019) found that 72% had experienced more than one patient's death by suicide, 15% more than six, and 3% had experienced more than ten patient suicides. Rothes et al. (2013) also found that male psychologists and psychiatrists had more patients' suicide than their female counterparts. The researchers did not explain this difference, however.

**Table 1. Literature review summaries.**

| Reference *Country* | Population and sample size | Emotional Impacts (most common) | Professional impacts (most common) | Exposure to suicide | Support sought | Mental Health outcomes |
|---|---|---|---|---|---|---|
| Murphy et al. (2019) *Ireland* | 179 mental health professionals | Sadness (79.5%, n = 65), shock (74.7%, n = 62) and surprise (68.7%, n = 57). Reactions lasted for < 6 months for most | Increased awareness of suicide risk (by nurses). Reduced professional confidence (66.7%), fear of negative publicity (54.2%), fear of litigation (49.4%). | 46.6% reported patient's suicide | Support through work (17.7%, n = 23), informal support (71.1%). Professionals more often preferred debriefing as a source of support. | Burnout 47.6%. |
| Awenat et al. (2017) *UK* | 20 mental health professionals | Guilt, fear of being blamed by patient's family and colleagues, frustration, and hopelessness on supporting repeat self-harmers | Reduced professional confidence | All reported a patient's suicide | Formal discussion of the experience, informal support from colleagues, friends, or family | Not reported |
| Gulfi et al. (2015) Switzerland | 713 mental health professionals | 49.6% felt responsible for the deceased | Professionals reported low to moderate reactions. | Professionals facing more than one patient suicide (Mean = 3.7, SD = 4.2) | 38.5% reported a need for social and/or psychological support. 38.8% sought support | Not reported |
| Dransart et al. (2015) Switzerland | 666 mental health professionals | 36.6% moderately impacted and 7.7% highly impacted | Not reported | Professionals facing more than one patient's suicide (Mean = 2.7, SD = 1.4) | 39.2% sought social and/or psychological support | Not reported |
| Fairman et a. (2014) USA | 186 mental health professionals | Guilt and self-doubt | Increased focus on suicide cues, enhanced care of patients | 32% reported a patient's suicide | Team-based support and debriefing most common coping strategies. Participants recommended facilitated debriefing, informal group support, and individual counseling | Not reported |
| Draper et al. (2014) *Australia* | 303 mental health professionals | Shock, sadness, anxiety, feeling upset, grief, anger, and guilt | Increased vigilance and awareness of suicide risk, more assessment and management of at-risk patients, increased referral to a psychiatrist. Sadness at work, loss of professional confidence | Not reported | Suicide-exposed professionals needed support or counseling more often than those exposed to other sudden deaths | Not reported |
| Dransart et al. (2014) *Switzerland* | 258 mental health professionals | Shock, helplessness, and sadness | Not reported | Not reported | Not reported | One in ten respondents above clinical cut-off for PTSD on the IES-R |
| Finlayson et al. (2019) *Australia* | 178 psychologists | Not reported | Not reported | 31.5% reported a patient suicide | Peer-support sought following poor workplace support | Not reported |
| Finlayson et al. (2018) *Australia* | 178 psychologists | Sadness, shock, and helplessness. | Caution with at-risk patients, increased attention to legal aspects, increased focus on suicide cues, increased peer consultation | 31.5% reported a patient suicide | Debriefing with colleagues most helpful coping strategy, followed by talking to supervisors. | Not reported |
| Darden et al. (2011) *USA* | 6 psychologists | Sadness | Questioned r clinical decisions, increased hypervigilance to suicide risk, concern over legal issues | Not reported | Discussing event was a coping strategy for recovery | Not reported |

*(Continued)*

**Table 1.** (Continued)

| Reference Country | Population and sample size | Emotional Impacts (most common) | Professional impacts (most common) | Exposure to suicide | Support sought | Mental Health outcomes |
|---|---|---|---|---|---|---|
| Gulfi et al. (2016) *Switzerland* | 271 psychologists and psychiatrists | Anxiety about working with at-risk patients and increased concern over professional competence | Increased focus on suicide cues, and hospitalization of patients, concern over legal matters, more consultation with supervisors and colleagues | Not reported | Not reported | Not reported |
| Wurst et al. (2013) *Germany* | 226 psychologists and psychiatrists | 39.6% of cases of suicide reported by professionals caused severe distress. Shock and sadness. Professionals involved in suicide case review reported higher levels anger and shame | 18.6% reported not being able to continue their work as usual | 72.1% of the sample have had experienced exposure to suicide. | 73.5% felt supported by employer and 87.9% by colleagues | Not reported |
| Leaune et al. (2019) France | 253 psychiatrists | Guilt and sadness were most frequently reported | 92% exposed to a suicide death modified their practice; 72% described this change as positive. 97.1% of those exposed to a serious suicide attempt modified their practice; 60% described this change as positive. | 49.4% of the sample was exposed to a patient suicide and 13.8% were exposed to a severe patient suicide attempt. | 25.6% of professionals exposed to suicide death reported feeling unsupported. Professionals commonly sought support from senior colleagues (56%), peers (38.4%), and informally (37.6%). 97.6% did not seek personal psychotherapy. | 16.8% of the participants scored moderate to high traumatic impact. 8.1% scored high to extremely high traumatic impact after a patient's suicide. |
| Gibbons et al. (2019) *UK* | 174 psychiatrists | Sadness (71%, n = 85), worry, anxiety, and fear (40%, n = 40), guilt and self-blame (31%, n = 36) | 98% reported effects on clinical practice. | 72% experienced >1 patient's death by suicide, 15% >6, and 3% had experienced >10. | Peer-support most helpful (48%, n = 43). Advice and support from senior clinicians (75%, n = 102) and formal support (70%, n = 97) were the supports most desired | 8% (n = 9) felt their symptoms met the clinical threshold for the diagnosis of a psychiatric disorder |
| Erlich et al. (2017) *USA* | 90 psychiatrists | Not reported | Half reported changing clinical practice, seeking supervision (50.9%, n = 27), using formal measures to assess suicidality (25%). 9.1% began using postvention protocols. 9.8% (n = 5) stopped accepting at-risk patients. | 66% had a client died by suicide. | They coped by reviewing their notes (81%, n = 47), informal supervision with a colleague (71.2%, n = 42), and discussion with a family member or friend (70.4%, n = 38) | Not reported |
| Rothes et al. (2013) *Belgium* | 107 psychiatrists | Suffering or distress (46.5%), sadness, despair and pain 25.6% felt impotent or powerless. 20% reported guilt and self-blame. 15% described feelings of anger, frustration, disappointment, misunderstanding towards patient, and 19% shock or disbelief | 45% reported changes in clinical practice. 54% of those reported more vigilant about suicide risk and increased accuracy in risk assessment and treatment. 22% improved practice by team conversation. 15% became more insecure toward suicide risk. 12% reported changes in relationship with other patients | 92% experienced a patient's suicide. Male professionals had more patients' suicide | 17% discussed case with team or colleagues | 92 reported possible sources of support including colleague consultation with team case review reported as most helpful |

(*Continued*)

**Table 1.** (Continued)

| Reference Country | Population and sample size | Emotional Impacts (most common) | Professional impacts (most common) | Exposure to suicide | Support sought | Mental Health outcomes |
|---|---|---|---|---|---|---|
| Scocco et al. (2012) Italy | 34 psychiatrists | Sadness, anger, disbelief, nervousness, and psychological pain. | In 66% of the cases, approach not affected. 18% reported improvement in approach, 3% reported a worsening in approach. 8% reported positive and negative consequences of approach. | 85% refer to a suicide attempt and 15% to a suicide death | 71% discussed event with peer colleagues, team, friends or relatives, patient's relatives, or senior colleagues. | Not reported |
| Kelleher et al. (2011) *Ireland* | 50 psychiatrists | 27.5% reported personal lives affected by sadness, low mood, and self-doubt | 32.5% professional life affected by sense of helplessness and reluctance to discharge patients. Increased awareness of suicide risk assessment and documentation | Not reported | 85% reported family support as helpful. 53% reported peer-support as helpful. | Not reported |
| Türkleş et al. (2018) *Turkey* | 33 nurses | Deep sorrow, anger, frustration, and blame by hospital staff | Increased awareness of suicide risk. | Not reported | Not reported | Not reported |
| Sherba et al. (2019) USA | 121 counselors and social workers | 19.83% (n = 24) Personal distress due to publicity. 44.5% (n = 53) personal distress at the possibility of litigation | 15.0% (n = 18) considering early retirement. 34.2% considering career change. 9.9% (n = 12) of all participants reported taking time off from work | 82,7% (n = 100) had a patient suicide or suicide attempt. | Team colleagues support (n = 114), family/partner support (n = 91), other clinical colleague support (n = 77), friends support (n = 72), other mental health professional support (n = 63). Team colleagues, other clinical colleagues, and other mental health professionals were the most helpful | Clinical risk for burnout moderate (9.2%, n = 11) and high (3.4%, n = 4). Risk for compassion fatigue moderate (12.5%, n = 15), high (16.7%, n = 20) and extremely high (13.3%, n = 16) |
| Hom et al. (2018) *USA* | 276 firefighters | exposure to suicide significantly associated with lifetime suicide attempts but not ideation or plans. Greater emotional impact associated with symptoms of depression, nightmares, insomnia and PTSD, and severe suicide risk | Not reported | 74.4% of exposure to suicide (personal life). 80.8% of the exposures occurred during their firefighting career | Not reported | Participants with lifetime exposure to suicide reported greater symptoms of depression, insomnia, and PTSD symptoms. Participants with career exposure to suicide reported greater symptoms of depression, nightmares, insomnia, and PTSD symptoms |
| Kimbrel et al. (2018) *USA* | 61 firefighters | Not reported | Not reported | All exposed to suicide; 34.8% via occupation; 23% most affected by occupational exposure. Mean of 13.1 exposures to suicide attempts and deaths throughout lifetime. | 13% of the firefighters sought treatment with 5% more than n once | 41% reported lifetime suicidal ideation, 11% in past year. 16% not affected by exposure to suicide; 8% reported a lifetime suicide plan; 12% positive screens for significant suicide risk. |

(*Continued*)

**Table 1.** (Continued)

| Reference Country | Population and sample size | Emotional Impacts (most common) | Professional impacts (most common) | Exposure to suicide | Support sought | Mental Health outcomes |
|---|---|---|---|---|---|---|
| Stanley et al. (2015) USA | 1027 firefighters | Not reported | Not reported | 92.4% had responded to a suicide attempt and 87.6% to a death by suicide. | Not reported | 46.8% reported suicide ideation, 19.2% suicide plan, 15.5% suicide attempts, and 16.4% non-suicidal self-injury. |
| Cerel et al. (2019) USA | 813 law enforcement | Not reported | Not reported | 95% exposed during work, mean 30.9 suicide scenes attended | Not reported | 14% identified moderate to severe depressive symptoms, 9.1% probable posttraumatic stress, 6.4% suicide ideation, 14.2% moderate to severe anxiety |
| Koch (2010) USA | 8 police officers | Coping strategies included: humor, depersonalization of the victim, faith, telling stories, blocking feelings, anger, reliance upon Police role, deep exploration of the deceased's life, engaging or disengaging from survivors, and hyper-alertness | Not reported | Not reported | Peer-support | Not reported |

*Authors, country in which the research was conducted, population and sample, summaries of the main findings regarding emotional impacts, professional impacts, exposure to suicide prevalence, support sought/suggested by the participants and mental health outcomes reported.

The most common emotional reaction reported amongst psychologists and psychiatrists was sadness, [4, 28, 29, 31, 34, 35, 37] with shock, guilt, self-doubt, low mood, psychological pain, and fear also being reported. Female psychiatrists and psychologists reported higher distress effects than males (n = 174) [29]. Another gender difference concerned the sense of responsibility for the death, with females reporting slightly higher scores on feeling responsible at the time of death (mean = 49.9, SD = 23.2) than males (mean = 38.8, SD = 25.5). Gender and responsibility for the deceased patient were significantly related to the emotional impact in [33]. Women professionals and those who felt more responsible for the death of the patient had more pronounced emotional reactions immediately after a patient's suicide. Research found that female psychiatrists reported significantly more feelings of failure and defeat (31%) than males (10%), however, no other difference was found regarding gender [4]. Wurst et al. (2013) found that women reported more emotional impacts than men, even though male professionals experienced more patient suicides. The gender differences presented in these papers are assumed without questioning why these differences were reported. This can reinforce a point of view that female psychiatrists and psychologists are more vulnerable to impacts than their male colleagues, reproducing gendered views. The results should be questioned, considering that male professionals might be less inclined to report emotional reactions, and therefore report lower levels of emotional impacts.

Overall, the most common professional reaction to exposure to suicide was increased awareness of suicide risk with patients. Other reactions included increased utilization of suicidality assessment tools, caution with at-risk patients, and increased attention to legal aspects of the therapeutic relationship. Peer-support and debriefing were the most prevalent support sought by these professionals after a patient's suicide.

No study reported suicide risk measures amongst the psychologists and psychiatrists exposed to suicide. Moreover, none of the studies of psychologists and psychiatrists evaluated the relationship between exposure to suicide and their own risk of suicidal thoughts and behavior. These findings highlight how existing research focuses on risk management and professional impacts, not personal and emotional ones. Therefore, it seems to disregard psychologists and psychiatrists as a potentially at- greater risk of suicide themselves.

## Nurses

Only one study was addressed to mental health nurses [38]. Nurses reported experiencing deep sorrow, anger, frustration, and feeling blamed by other staff following a patient's suicide. Some nurses also described feeling angry and resentful if they perceived the suicide attempt as attention-seeking [38]. Many nurses felt anxiety, stress, unrest, and fear as a result of feeling responsible for failing to prevent the suicide. The most prevalent professional change was the nurses' increased awareness of suicide risk amongst patients.

## Counselors and social workers

The only paper reporting the impacts of exposure to suicide among counselors and social workers (USA) reported that 82.7% of the sample had a patient attempt or complete suicide [39]. The emotional impacts reported were related to the press publicity of the suicide and fear of litigation, with 19.8% reporting distress after the suicide press publicity and 44.5% of the professionals experiencing distress at the possibility of litigation. This led to professionals considering changing careers (34.2%) and retiring early (15%). Peer-support was the type of support most sought and most often described as helpful. Burnout and compassion fatigue were outcomes after a exposure to suicide, with more than 42.5% of the suicide affected scoring between moderate and extremely high for compassion [39]. These findings are relevant given the fact that these professionals may have less training in mental health and suicide prevention in comparison to other professional categories.

## Firefighters

Three papers examined exposure to suicide amongst firefighters, all quantitative and from the USA. The sample size of the studies ranged from 61 to 1,027, and all used web-based surveys. The prevalence of exposure to suicide while on-duty (including suicidal behaviors) ranged from 34.8% to 92.4% [40–42]. Hom, Stanley [40] report the prevalence of exposure to suicide in the firefighters' personal life (74%), rather than their exposure while on duty. For those who knew someone who died by suicide, 54% reported the person who died was another firefighter (but not a close colleague) and 31.1% reported it was a fellow firefighter (a close professional colleague). For those who reported exposure to suicide during their personal life, 39% reported being "affected by suicide" but not to the extent that it "disrupted life". Two studies [41, 42] did not report the type of emotional reactions experienced by the firefighters. Hom, Stanley [40] are the only authors to report data regarding emotional reactions. They found that a greater emotional impact from exposure to suicide was associated with a statistically significant higher risk of lifetime suicide attempts, suicide risk, depressive symptoms, nightmares, insomnia, and more severe PTSD symptoms.

Likewise, Kimbrel et al (2016) reported that suicidal behavior or the death of another firefighter or co-worker was common amongst firefighters and significantly impacted them. However, only the on-duty exposure to suicide was correlated to increased suicide risk in these firefighters. Therefore, the research did not explore the impacts of a firefighter colleague's suicide.

In terms of the link between exposure to suicide and risk of suicidal behavior, Stanley et al (2016) reported that firefighters who attended a suicide-related call were more likely to report suicidal ideation and attempts. No paper reported the professional impacts of exposure to suicide on firefighters.

## Police officers

Two papers, both from the USA, were focused on police, one qualitative (n = 8), and one collecting data through web-survey (n = 813) (Cerel et al., 2019; Koch, 2010). Only one paper reported the prevalence of exposure to suicide [3], with 95% of Law Enforcement Officers (LEOs) reporting encountering suicide while on-duty. In this study, 70.7% of the sample reported high exposure to suicide (defined as being exposed to 10 or more suicides in their entire career). Furthermore, 73.4% reported knowing someone from their personal life who died by suicide. The higher the level of exposure to suicides on duty, the more likely it was that officers reported a PTSD diagnosis, suggesting that attending suicide scenes has a particular PTSD impact.

Koch (2010) categorized strategies police officers reported coping with the impact of a suicide scene. The research, however, did not explore all the extensions of these strategies, as they can be adaptive or maladaptive, thus leading to changed risks for mental health outcomes. The in-depth exploration of these strategies would be valuable to understand and design interventions.

## Discussion

This narrative scoping review has documented and synthesized the current state of research on the impacts of work-related exposure to suicide on MHPS and first responders as professionals in these occupations are more likely to encounter suicide and suicidal behavior than the general population.

Our review has shown that 17 out of the 25 papers used web-surveys as their method of data collection. While this method is growing in popularity, there are two key methodological limitations to its use [44]. First, unless a specific group has been sampled and the web survey distributed to individuals, then it is not possible to know the population the survey has reached. Without this information, researchers cannot be sure that the sample who have responded to the survey are representative of the population. In other words, the research findings cannot be generalized. Second, web-surveys are prone to selection bias, as respondents may select themselves into the sample. For example, those who are more interested in the subject, are health literate and have access to the internet are more likely to complete the web survey than those who are not. In the 17 studies that have used web surveys described in this review, the researchers have been able to describe the population sampled since they have included specific occupational/professional group (e.g., police officers), however self-selection remains a limitation to the generalization of the findings. This issue should be addressed in future research.

Police officers have the highest on-duty exposure to suicide, followed by firefighters, then psychiatrists, with counselors, social workers, and psychologists having the lowest exposure. No research reported patient exposure to suicide among nurses. Most of the studies were conducted in the USA (n = 7), while the other studies were conducted in the UK, Australia, Ireland, France, Belgium, Italy with no research from other countries All the papers that assessed exposure to suicide used web-surveys.

The impacts of exposure to suicide were reported in personal and professional domains for most groups. Personal impacts were examined, focused on emotional reactions, especially

shock and sadness. The highest prevalence of shock and sadness was reported among MHP [4, 21, 29]. The lack of reporting on the full range of emotional impacts can be explained by a lack of qualitative research in this field which may be better suited to investigating this phenomenon due to its subjective nature. It is also noteworthy, that most of the research in this review that assessed emotional impacts used the Acute Emotional Impact Scale or an adaptation of this scale, restricting participants to reporting only the emotional items made possible in the scale (i.e. shock, sadness, guilt). A much wider range of emotional responses might be experienced, such as frustration, angst, and others, but the utilization of scales restricts the reporting and expression of such feelings.

Professional reactions described showed that increased awareness of suicide risk and suicide cues were the most common impacts on professional practice. No paper reported professional reactions among nurses, firefighters, and police, even though we anecdotally expect that nurses might change their practice after a patient's suicide.

Qualitative studies tended to explore the coping strategies professionals used. Peer-support, although it was never defined or described, was the most common coping strategy used by all professionals, without differences between first responders and MHP.

Qualitative studies also showed that mental health professionals have an increased fear of litigation after a patient's suicide. The critical incident review of the safety procedures taken by the professionals prior to the suicide has been reported to be a distressing event, potentially exacerbating the impacts of the exposure to suicide itself [16].

An interesting finding arising from the review is the lack of mental health and suicidality assessment of the mental health professionals themselves. No article discussed suicidality amongst mental health professionals, with PTSD and burnout being the only mental health outcomes explored. The lack of data regarding mental health outcomes makes it impossible to compare exposure to suicide impacts on mental health professionals with those on first responders. We do not have data to indicate which profession is most impacted by exposure to suicide. The lack of studies aimed at mental health outcomes after exposure to suicide also suggests that suicide researchers may not pay enough attention to suicide risk among health professionals, assuming that, as they are trained to give support in these situations, they might be better equipped to manage their emotions. However, mental health care professionals are among the occupations with increased suicide risk [5]. The recent review by Dutheil, Aubert [5] has highlighted the lack of research examining the etiology and the transition from suicide ideation to suicide attempts among health professionals. Therefore, we suggest exposure to suicide should be considered as a risk factor with more research needed to examine the link between exposure and suicidality amongst MHP and first responders.

This scoping review highlights several gaps in the research base on exposure to suicide in mental health professionals and first responders. First, there was a complete lack of research on exposure to suicide amongst paramedics and ambulance workers. Two reviews suggested that attending suicide is an emotional burden, no papers were focusing solely on exposure to suicide [12]. Furthermore, there were only two studies examining police and three examining firefighters, all from the USA. There is a need for research addressing the issue of exposure to suicide among these professionals and, that focus on its emotional and professional impacts.

Second, there is a lack of use of validated scales for assessing emotional impacts. Similar to the review of Séguin, Bordeleau [45], the papers in this review that used the Event Impact Scale or a sub-scale of it also used the instrument to set cut-offs scores to categorize clinical levels of distress or as a proxy measure to PTSD [23, 24, 27, 28, 33]. Following the discussion of Séguin, Bordeleau [45], the authors of the scale do not report any cut-off score as to categorize, therefore, the usage of the scale can lead to an inflate result of professionals being clinically

distressed. Future research should use properly validated scales to evaluate emotional distress to identify those that are clinically affected and might benefit from support.

Third, our review highlights the lack of qualitative research in this area. While quantitative research enables hypothesis testing, quantitative assessment of the phenomenon and the correlations, qualitative in-depth investigations of the impacts of exposure to suicide are necessary to better understand how individuals subjectively make meaning of this experience. Questions remain, for example around what factors prevent professionals from seeking support, how the event affects their willingness to work with at-risk patients, and other concerns related to how the event impacts the professionals.

The fourth insight brought by the review is the lack of theory informing the research in this area. Only two studies [3, 40] utilized any theory to inform their inquiry or interpretation of data; Joiner's Interpersonal theory of suicide (IPTS). This theory hypothesizes that the wish for suicide emerges from two psychological constructs–perceived burdensomeness (feeling like a burden to friends and loved ones) and thwarted belongingness (a sense of low belongingness or social alienation [46]. Joiner believes that individuals acquire the capability for suicide through a process of repeatedly experiencing painful events such as workplace exposure to suicide. The outcome of this repeated exposure to such events is the loss of the instinctual fear of death or what has often described as survival instinct. According to IPTS both direct and vicarious exposure can heighten the capability for suicide [47]. Two papers listed on our review have used the IPTS theoretical framework, utilizing the Acquired Capacity for Suicide Scale (ACSS) and the Interpersonal Needs Questionnaire (INQ) to access the theory's constructs comparing it with the suicidality measures, such as the Suicide Behavior Questionnaire Revised SBQ-R [3, 40]. The results showed IPTS was not tested among mental health professionals' exposure to suicide. Although they used measures derived from IPTS, no research compared the theory assumption of exposure to suicide as a possible cause of suicidality. Theoretically informed research, which focuses on understanding and not just explanation, is needed to advance our understanding of the context and risk of workplace exposure to suicide [48]. This in turn will help to inform the development of targeted evidence-based health promotion interventions in these occupations.

Our review points out the high impacts exposure to suicide has on different MHPs, with outcomes to the professionals' mental health and well-being. Those outcomes, such as PTSD and anxiety are correlated with suicide risk [1]. Due to the lack of research on first responders and no research among paramedics and ambulance personnel, we can not draw conclusions regarding the impacts of exposure to suicide among these professionals. Moreover, exposure to suicide seems to be related to suicide risk among MHPs and first responders and should be the focus for suicide prevention protocol.

## Strengths and limitations

This scoping review has several strengths. First, to our knowledge, this is the first review to examine the impacts of exposure to suicide framed in terms of occupational exposure. It is also the first to review the literature on the impacts of exposure to suicide on first responders. Second, our review involved a systematic and replicable search for all major databases. Third, the papers included in our review were evaluated for quality, enabling us to identify the strengths and weaknesses of the research and its reporting. A fourth limitation identified in the literature is the lack of description on the relationship of the health professionals and the patient who died by suicide (e.g.: case manager, psychotherapist, or prescriber). This kind of refinement on the data gathering can elucidate how relationship to the patient can influence on professionals and personal reactions.

Like any review, there are some limitations. Only peer-reviewed studies published in English in the last 10 years were eligible for inclusion. It is possible that relevant older studies, non-English studies, or grey literature might provide additional information.

## Conclusion

The scientific body of literature has focused on studying exposure to suicide among different health professionals, but less attention has been given to first responders, albeit with some limited, mainly quantitative research. Our review highlights some of the impacts exposure to suicide has on MHPs and how this can contribute to adverse mental health outcomes that are related to increased risk for suicide. The lack of research among first responders means we are unable draw any conclusions on the impacts of exposure to suicide and suicide risk in these groups, this is especially the case for paramedics. There is a need for more research on this phenomenon amongst police, firefighters, and ambulance workers/paramedics specifically, considering the fact that attending suicide scenes can increase PTSD outcomes. Further, more in-depth exploration of the impacts of exposure to suicide among MHP and first responders is necessary to understand how exposure to suicide through work impacts on personal and professional life, how it is experienced and managed, links with risks of suicide in these groups, and how this and other emotional distress might be reduced.

## Supporting information

**S1 Checklist. Preferred Reporting Items for Systematic reviews and Meta-Analyses extension for Scoping Reviews (PRISMA-ScR) checklist.**
(PDF)

**S1 Appendix. Complete research strategy.**
(DOCX)

## Author Contributions

**Conceptualization:** Renan Lopes de Lyra, Sarah K. McKenzie, Susanna Every-Palmer, Gabrielle Jenkin.

**Investigation:** Renan Lopes de Lyra.

**Methodology:** Renan Lopes de Lyra, Sarah K. McKenzie, Gabrielle Jenkin.

**Writing – original draft:** Renan Lopes de Lyra.

**Writing – review & editing:** Sarah K. McKenzie, Susanna Every-Palmer, Gabrielle Jenkin.

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
