## [Decision Letter · Decision Letter 0]

6 Mar 2021

PONE-D-20-36231

Suicide exposure through work: a scoping review of research on mental health professionals and first responders

PLOS ONE

Dear Dr. Lopes De Lyra,

Thank you for submitting your manuscript to PLOS ONE. After careful consideration, we feel that it has merit but does not fully meet PLOS ONE’s publication criteria as it currently stands. Therefore, we invite you to submit a revised version of the manuscript that addresses the points raised during the review process.

We look forward to receiving your revised manuscript.

Kind regards,

Vedat Sar, M.D.

Academic Editor

PLOS ONE

Journal Requirements:

"No"

Reviewers' comments:

Reviewer's Responses to Questions

**Comments to the Author**

1. Is the manuscript technically sound, and do the data support the conclusions?

Reviewer #1: Yes

Reviewer #2: Yes

2. Has the statistical analysis been performed appropriately and rigorously? 

Reviewer #1: N/A

Reviewer #2: I Don't Know

3. Have the authors made all data underlying the findings in their manuscript fully available?

Reviewer #1: Yes

Reviewer #2: Yes

4. Is the manuscript presented in an intelligible fashion and written in standard English?

Reviewer #1: Yes

Reviewer #2: Yes

5. Review Comments to the Author

Reviewer #1: This is an interesting and important review of the literature on mental health and first responder exposure to suicide and the impact on them. The conclusions are supported by the review and are compelling.

There were no statistics done but they do comment on the methodology of the studies they review.

The biggest issue is the wordiness and size of "table 1. Literature review summaries." Point form rather than sentence form would be more appropriate.

A few minor points.

Page 5, line 4, please define "high emotional labor".

Page 7 line 1 and 2 are not necessary about why the spreadsheet was turned into a table.

Page 8, first sentence, 15+5+6 does not equal 25.

Page 14, 6th column under Erlich et al, the authors forgot to replace "commit" suicide with "died by" suicide.

Page 21, first sentence is confusing about why the analysis wasn't broken down by professional category and whether it was this paper's authors or the authors of the papers they reviewed who did this. It would make sense if it read " The MHPs group includes psychologists, psychiatrists, social workers and nurses."

Page 22, last paragraph needs clean up of the text citations.

Page 23, line 6, 7 and 8. What evidence do the authors have that male professionals are less inclined to report emotional reactions? If there is no proof, this clause of the sentence should be more tentative.

Page 25 , last line, "attending suicide scenes have a particular PTSD impact". The verb should be has. Furthermore, I think this is a very important point and should be re-iterated in the conclusion.

Page 26, line 14 & 15, "All the papers that assessed suicide exposure used web-surveys." I was surprised to find out that most of the methodology of the reviewed studies was web surveys, in the discussion. Perhaps there should be a column in the table 1 for methodological design. A quick count revealed that 17/25 studies were web based. Is this true? This also merits some discussion of the shortcomings of web based surveys.

Page 27, last paragraph, I really like the conclusion that "suicide researchers may not pay enough attention to suicide risk among health professionals, assuming that, as they are trained to give support in these situations, they might be better equipped to manage their emotions."

Page 28, line 17 should read "usage of the scale can lead to an inflated result of professionals being clinically distressed."

Page 30, last line. should read "might be reduced".

Reviewer #2: I have a few suggestions to make about clarity of phrasing:

The term "suicide exposure" isconvenient but not as telling as "exposure to suicide"

I don't understand the meaning of "scoping" review

Consider writing in the title "...review of research on the experience of mental health professionals and first responders"

Was it possible to know differences in the nature of the relationship with the client among mental health professionals, e.g., case manager, prescriber, psychotherapist, etc. It should be important to include.

I find important the "insight" in the Discussion about lack of theory in "framing" the research. I suggest saying more about what "framing" means and what hose twotheories say.

6. PLOS authors have the option to publish the peer review history of their article (what does this mean?). If published, this will include your full peer review and any attached files.

Reviewer #1: No

Reviewer #2: No

---

## [Author Response · Author response to Decision Letter 0]

1 Apr 2021

Thank you very much for your favorable assessment of our manuscript and for

reading it so carefully and making such constructive recommendations. All recommendation were accepted and included on the text. We feel the recommendations have greatly improved the quality and clarity of

our manuscript. 

Journal recommendations

We have corrected the style elements in line with PLOS requirements.

Funding information was corrected.

Competing interests was corrected.

Caption for the supporting file was corrected.

Reviewer one

Thank you for your review and suggestions to improve our work.

The biggest issue is the wordiness and size of "table 1. Literature review summaries." Point form rather than sentence form would be more appropriate.

- Table 1 was revised and reworded.

Page 5, line 4, please define "high emotional labor". 

– Emotional Labor defined.

Page 7 line 1 and 2 are not necessary about why the spreadsheet was turned into a table. 

– Phrase removed.

Page 8, first sentence, 15+5+6 does not equal 25. – We have recounted and noticed that it was one extra paper that was not included in the final version. The numbers were corrected.

Page 14, 6th column under Erlich et al, the authors forgot to replace "commit" suicide with "died by" suicide. 

– Replaced.

Page 21, first sentence is confusing about why the analysis wasn't broken down by professional category and whether it was this paper's authors or the authors of the papers they reviewed who did this. It would make sense if it read " The MHPs group includes psychologists, psychiatrists, social workers and nurses." 

– We have reworded the phrase in order to make sense that the authors of the cited papers have included all those professionals as a single group during analyses.

Page 22, last paragraph needs cleanup of the text citations. 

– The last paragraph was cleaned, and the authors were excluded, only the reference number left.

Page 23, line 6, 7 and 8. What evidence do the authors have that male professionals are less inclined to report emotional reactions? If there is no proof, this clause of the sentence should be more tentative.

-After discussion, we decided to remove the phrase as it was not supported by the data, but a anecdotical conclusion.

Page 25 , last line, "attending suicide scenes have a particular PTSD impact". The verb should be has. Furthermore, I think this is a very important point and should be re-iterated in the conclusion.

-The verb was corrected and a few lines were included to the conclusion highlighting this aspect.

Page 26, line 14 & 15, "All the papers that assessed suicide exposure used web-surveys." I was surprised to find out that most of the methodology of the reviewed studies was web surveys, in the discussion. Perhaps there should be a column in the table 1 for methodological design. A quick count revealed that 17/25 studies were web based. Is this true? This also merits some discussion of the shortcomings of web-based surveys.

- We have counted and confirmed this number. A few lines were written to address the potential issues related to this fact, highlighting the limitations of the use. 

Page 28, line 17 should read "usage of the scale can lead to an inflated result of professionals being clinically distressed."

-Phrase corrected.

Page 30, last line. should read "might be reduced".

-Corrected.

Reviewer two

The term "suicide exposure" is convenient but not as telling as "exposure to suicide".

-We have replaced all the entries of “suicide exposure” for “exposure to suicide” on the text.

I don't understand the meaning of "scoping" review

Consider writing in the title "...review of research on the experience of mental health professionals and first responders"

- After discussion we decided to change the tittle. Although we explain that our review fits in the description of a scoping review on the method section.

Was it possible to know differences in the nature of the relationship with the client among mental health professionals, e.g., case manager, prescriber, psychotherapist, etc. It should be important to include.

-We came back to the papers, but they did not describe the nature of relationship of professionals. This may be a limitation on data, and we have addressed this issue on our discussion. Thank you.

I find important the "insight" in the Discussion about lack of theory in "framing" the research. I suggest saying more about what "framing" means and what those two theories say.

-We have written some lines trying to describe how research that used theory framing should work, trying to link exposure to suicide and the suicide risk among professionals.

Thank you for your revision.

---

## [Decision Letter · Decision Letter 1]

19 Apr 2021

Occupational exposure to suicide: A review of research on the experiences of mental health professionals and first responders

PONE-D-20-36231R1

Dear Dr. De Lyra,

We’re pleased to inform you that your manuscript has been judged scientifically suitable for publication and will be formally accepted for publication once it meets all outstanding technical requirements.

Kind regards,

Vedat Sar, M.D.

Academic Editor

PLOS ONE

Additional Editor Comments (optional):

Reviewers' comments:

Reviewer's Responses to Questions

**Comments to the Author**

1. If the authors have adequately addressed your comments raised in a previous round of review and you feel that this manuscript is now acceptable for publication, you may indicate that here to bypass the “Comments to the Author” section, enter your conflict of interest statement in the “Confidential to Editor” section, and submit your "Accept" recommendation.

Reviewer #1: All comments have been addressed

Reviewer #2: All comments have been addressed

2. Is the manuscript technically sound, and do the data support the conclusions?

Reviewer #1: Yes

Reviewer #2: Yes

3. Has the statistical analysis been performed appropriately and rigorously? 

Reviewer #1: N/A

Reviewer #2: I Don't Know

4. Have the authors made all data underlying the findings in their manuscript fully available?

Reviewer #1: Yes

Reviewer #2: Yes

5. Is the manuscript presented in an intelligible fashion and written in standard English?

Reviewer #1: Yes

Reviewer #2: Yes

6. Review Comments to the Author

Reviewer #1: Thank you for making all the requested changes. I am glad the authors feel the reviewer comments enhanced the paper

Reviewer #2: The authors have responded constructively and sufficiently to all comments and suggestions from the initial review.

7. PLOS authors have the option to publish the peer review history of their article (what does this mean?). If published, this will include your full peer review and any attached files.

Reviewer #1: No

Reviewer #2: No

---

## [Editor Report · Acceptance letter]

21 Apr 2021

PONE-D-20-36231R1 

Occupational exposure to suicide: A review of research on the experiences of mental health professionals and first responders 

Dear Dr. Lyra:

I'm pleased to inform you that your manuscript has been deemed suitable for publication in PLOS ONE. Congratulations! Your manuscript is now with our production department. 

Kind regards, 

on behalf of

Dr. Vedat Sar 

Academic Editor

PLOS ONE